# OpenReview forum: "Atomos: Hierarchical Reasoning From Atomic Steps"
_ICLR.cc/2026/Conference — Submitted to ICLR 2026_

### Official Review · Reviewer_URmB · 2025-10-23

**Soundness:** 2
**Presentation:** 1
**Contribution:** 1
**Rating:** 0
**Confidence:** 4

**Summary:**

This paper proposes a hierarchical reasoning pipeline to let the LLM smarty allocate the computation in parallel sampling and step-level self-verification. Theoretical results justify its soundness and the paper qualitatively verify the proposed method on one IMO problem.

**Strengths:**

1. This paper has made several interesting analyses in the LLM reasoning behaviors under certain assumptions, which could be useful takeaways if the assumptions are well-hold in practice
2. Atoms reach a correct answer for IMO2025 P6 (claimed by authors, I do not have the knowledge to verify whether the answer is true) simply based on existing models, which is impressive.

**Weaknesses:**

This paper is not really presented in a scientific way.
- First, the method part is very vague, it remains unclear to me what is the key component of the proposed method. No hyper parameters are discussed.
- Second, the empirical part only contains a qualitative result on 2025IMO P6, with only Gemini-2.5-Pro as the base model. There is **no quantitative results at all**, it is almost impossible to draw any conclusions.
- Third, the readability is also a big issue, this paper is full of self-defined terms, lacking connections to previous methods.
- Finally, all the so-called laws discussed in this paper are based on strong assumptions, without any careful validation on real-world data. For example, it is impossible to know the failure rate of a step from a certain LLM. From my point of view, these laws do not really contribute any theoretical value as well, then in that case, verifying with solid empirical observations are important.

**Questions:**

I do not have specific questions, I think this paper should be re-written in a more research style.

---

### Official Review · Reviewer_swQb · 2025-10-30

**Soundness:** 2
**Presentation:** 1
**Contribution:** 1
**Rating:** 0
**Confidence:** 3

**Summary:**

The work proposes to perform reasoning in LLMs through a set of reasoning chains rather than a single Chain of Though (CoT). In contrast to CoT, a plan is made in advance to obtain self-contained steps. These steps are each generated and verified using the LLM (a self-checking loop: propose, verify, retry) to increase robustness of each chain. As a main contribution (line 861), the work proposes a compute law and scheduler: a rule that states how much compute to allocate to generating multiple pathways versus -per pathway. This approach is evaluated on IMO2025 P6, a specific problem of the International Mathematical Olympiad 2025.

**Strengths:**

The work aims to address a relevant problem: more robust reasoning in LLMs.

The proposed solution is simple: first generate a plan of atomic task units, then solve and verify each task, balancing verification and re-generation of a task with the generation of more proving pathways. The work also contains two additional proposed computational laws.

**Weaknesses:**

**experiments**

The approach is demonstrated on a single reasoning problem instance. Furthermore, the main contribution, the compute allocation laws, is not discussed within this problem instance. It is unclear what compute went into what phase and if suboptimal results were more likely to be achieved otherwise. I.e., the proposed 'laws' are not tested.

Detailed proof of IMO P6 in appendix is unclear. It starts with "Input:", which then seems to include not only the problem statement but also a problem analysis, case analysis, induction and derivation, ... final conclusion; and it ends with "Now you start thinking about how to prove the conclusion". So it appears that the LLM receives much more than simply the problem statement. Furthermore, there appears to be user interaction as opposed to autonomous reasoning? Related, how do we come up with `` use at least **10** methods, and give **100** solutions to the subpath nodes'' (line 1372-1374)?

**presentation**

The presentation is very pompous. E.g., "trinity of systemic flaws", "genesis of hallucinations", "systematically dismantles the sources of unreliability", "operational proxy", "profound insight into the economics of reliability", "from a probabilistic gamble to a predictable science of compute allocation".

The presentation is overly verbose and the space could be used for a more thorough explanation of the contribution. E.g., section 2.1, could easily be summarized in 3 sentences; Eq 1 and 2 are unnecessary and obvious. Sect 2.2 and 2.4 could similarly be summarized. The practical usefulness of Eq 3 is not clear to me, since the "model-specific cognitive threshold" A_max is unquantiable and unknown. I would also guess that such a threshold should depend on the topic, etc., and would not be a single value for the entire model. In contrast, a more detailed related work section is moved into the appendix instead of the main text.

It is unclear how to deal with a dependency between the results of units (line 881). E.g., the formula hypothesis of one atomic unit that is used in the next. How did it get to this hypothesis formula without actually executing the first atomic unit before planning the remainder. The related work states that the Tree and graph-structured prompting approaches are brittle to cascading errors in serial generation, but when there are dependencies Atomos has the same problem? In case there is no dependence between the units, as in they are really atomic, then the approach becomes equivalent to verification after chain generation?

Several times symbols are used without explicitly defining them. For example, line 107 C_p^* and C_w^*, line 264 c_{ver}, c_{gen}. the reader must induce this from the rest of the text. Also, if symbols are not used elsewhere it may not be necessary to introduce them (i.e., epsilon on line 301 is not used afterwards).

**novelty**

The compute laws and scheduler are stated to be the main contribution (line 861).

The first law deals with an optimal time allocation for budget C between exploring more chains (C_w) and improving their reliability (C_p). The law states that the optimal allocation is given by $C_p = \alpha C$; $C_w = (1-\alpha) C$. This is unclear to me. It seems that $\alpha$ is either defined as the optimal fraction already, in which the case law is obviously following from that definition and is not useful, or $\alpha$ has a different meaning in which case how to obtain the law is insufficiently explained.

The second law I could not verify, it skips to the conclusion too fast. It starts with $\delta$, that is defined as global budget, such that $\delta/N_s$ is the budget you have per step to ensure no failure. This is then repurposed as a probability "with a global failure probability not exceeding $\delta$", and also becomes a reliability requirement "the reliability requirement $1/\delta$.

**Questions:**

## Questions

How are results of each world combined (line 895)?

Line 398 mentions a reasoning graph. From previous text I concluded a set of separate pathways (chains)) that are never merged into a more general tree/DAG, is that correct?

"Complex problems can be decomposed into verifiable atomic units whose unitary difficulty stays within the model's reliable operating regime". [...] "This enforced decomposition guarantees that the model is never asked to perform a cognitive jump it cannot reliably make." How does Atomos guarantee that this is indeed the case, since it knows neither A_max, nor C_u(s_i)?

Table 3 assigns a Conceptual Leap as the cause of the baseline's failure. However, a conceptual leap relates to a model's cognitive threshold. It is unclear to me why we are allowed to conclude a conceptual leap from a single failed execution.

Table 3 shows Atomos Trajectory. The Plan already contained a hypothesis formula. How is this possible without first solving previous tasks?


## Remarks

I assume the LLM is used as the verifier; this seems not to be mentioned explicitly except for in the appendix (line 854).

The critique generated by the verification step is not used in the next generation step (line 901, 904).

In the abstract, what exactly is "extreme reliability" as opposed to just reliability?

Figure 2 caption mentions a,b,c but there are no a,b,c in the figure.

Why 'laws' instead of 'theorems'?



Typos:
* "stepsan"
* "goal-seekingthat
* "flawsbrittle"
* "atomic"meaning
* "authorityclear"
* "** Self-Checking Loop: **"

---

### Official Review · Reviewer_HCyN · 2025-11-01

**Soundness:** 2
**Presentation:** 3
**Contribution:** 3
**Rating:** 4
**Confidence:** 3

**Summary:**

This paper addresses the critical problem of unreliability in large language model (LLM) reasoning, which suffers from cascading errors in multi-step tasks and a bias towards probabilistic, "hasty" shortcuts. The authors introduce Atomos, a training-free, test-time framework that decomposes complex problems into a graph of "atomic steps." The core mechanism is a low-overhead Propose-Verify-Retry loop where the same base model acts as its own verifier, leveraging a "verification asymmetry" (i.e., verifying an atomic step is much cheaper and more reliable than generating it).

The authors formalize this approach with two "Reliability Laws," which provide a theory for optimally allocating compute between exploration breadth ("world sampling") and verification depth ("path sampling"). A key theoretical claim is that the cost of achieving high reliability scales only linearly with task complexity but polylogarithmically with the reliability requirement, making extreme reliability "surprisingly affordable."

**Strengths:**

Novelty: The "verification asymmetry" insight is powerful. Using the same base model to verify its own atomic steps, rather than requiring a separate, stronger verifier model or expensive fine-tuning, is an elegant and practical approach.

Theoretical Contribution: The formalization of the compute-reliability trade-off into two "Reliability Laws" is a significant contribution. Law 2, which posits a polylogarithmic cost for increasing reliability, is a particularly strong and optimistic finding for the field.

Clarity of Diagnosis: The paper provides a clear and useful vocabulary for well-known failure modes, such as the "Uncontrollable Conceptual Leap" (a step's complexity exceeding the model's reliable limit) and "Hasty Goal-Seeking" (bias for fluency over validity).

Impressive Empirical Demonstration: While only a single data point, the autonomous solution to a complex IMO problem is a highly compelling "grand challenge" result that effectively showcases the potential of the Atomos framework.

**Weaknesses:**

Lack of Broad Empirical Validation: The paper's empirical evidence rests entirely on a single N=1 case study (the IMO problem). This makes it impossible to assess the generality of the framework or the "Reliability Laws." The paper's abstract claims "predictable accuracy-compute trade-offs across benchmarks," but no benchmark results (e.g., on MATH, GSM8k, etc.) are presented, which is a significant omission.

The "Brittle" Planning Phase: The framework relies on an initial "explicit planning phase" to decompose the problem into atomic steps. However, as the authors admit in the "Limitations" section, this planning phase is itself not verified. This is a critical weakness, as it suggests Atomos may simply push the "brittle chain" problem up one level, from execution to planning. A flawed plan, even if executed with perfect reliability, will lead to a wrong answer.

Uninterrogated Verifier Accuracy: The framework is built on the assumption that the base model is a reliable verifier for atomic steps. This is a strong assumption. What if the verifier is overconfident and accepts a flawed step, or under-confident and rejects a correct one? The paper does not provide an empirical analysis of the verifier's own accuracy or how the system handles verifier-induced errors.

**Questions:**

The IMO result is striking, but to validate the generality of the "Reliability Laws," could the authors provide results on standard reasoning benchmarks (like MATH, AIME2024, AIME2025)? Specifically, do the predicted optimal compute allocations from Law 1 hold true on these benchmarks?

The unverified planning phase seems to be the framework's Achilles' heel. How does the model guard against the planner itself taking a "hasty goal-seeking" shortcut? What is the measured failure rate of the planner, and how does this affect the system's end-to-end reliability?

Could the authors provide a more concrete cost analysis? For example, on a representative problem from the MATH, AIME2024, AIME2025 benchmark, what is the actual token/compute overhead of Atomos required to match the accuracy of a well-tuned, best-of-N Chain-of-Thought baseline?

---

### Official Review · Reviewer_i9hU · 2025-11-03

**Soundness:** 2
**Presentation:** 2
**Contribution:** 2
**Rating:** 2
**Confidence:** 2

**Summary:**

The paper proposes ATOMOS, which is a training-free inference engine that works by executing a graph of minimal self-verifying atomic units. The authors provide a series of definitions to justify their method, then proceed to show that ATOMOS with Gemini is able to prove IMO 2025 problem 6.

**Strengths:**

- The ATOMOS engine is able to solve IMO P6, which I find extremely impressive. If the method were true and generalizable, then this would indeed be a very strong result.

**Weaknesses:**

- No actual quantitative results. No experiments. Only 1 case study example.
    - The entire paper only had 1 empirical example, which was the IMO P6 example. Because of that, I'm not sure how widely this method can be applied.
    - The authors also didn't try running it on other models like Qwen or GPT, which adds to the questions about its generalizability.
- I found the paper very hard to understand. Not sure if I was just lacking context, but it seemed too overly theoretical to me. In my opinion I think it's clearer for readers if the main body contained experiments, then the theoretical explanations were moved to the appendix. This paper did the opposite, where the main body was full of definitions and conceptual explanations.

**Questions:**

- Figures are a bit unclear. For example, I found it hard to understand what was happening in Figure 1. Maybe some descriptions in the caption would help
- Random thing I wanted to flag (not sure if this is relevant; feel free to ignore) -- For some reason I felt like this paper felt more "squished" than the other papers I was reviewing. I believe the authors changed the spacing a bit. Not familiar with ICLR spacing rules, so maybe this is just totally fine.

---

### Meta-Review · Area_Chair_tUae · 2026-01-07

**Summary:**

This submission introduces ATOMOS, a training-free test-time framework that decomposes reasoning into atomic, self-verifiable units and argues for principled compute allocation between exploring multiple solution paths and increasing per-step reliability. Reviewers agree the core idea is interesting but they have several concerns and no rebuttal was presented.

As a result, we will have to reject the paper.

**Reviewer Concerns:**

No concerns were addressed by the rebuttal since there's no rebuttal.

**Reviewer Scores:**

All scores will remain the same given that no rebuttal was provided

---

### Decision · Program_Chairs · 2026-01-26

Reject